# Mathematical Modeling Support for Lung Cancer Therapy—A Short Review

**DOI:** 10.3390/ijms241914516

**Published:** 2023-09-25

**Authors:** Jaroslaw Smieja

**Affiliations:** Department of Systems Biology and Engineering, Silesian University of Technology, ul. Akademicka 16, 44-100 Gliwice, Poland; jaroslaw.smieja@polsl.pl

**Keywords:** mathematical modeling, lung cancer, anticancer therapies, metastasis

## Abstract

The paper presents a review of models that can be used to describe dynamics of lung cancer growth and its response to treatment at both cell population and intracellular processes levels. To address the latter, models of signaling pathways associated with cellular responses to treatment are overviewed. First, treatment options for lung cancer are discussed, and main signaling pathways and regulatory networks are briefly reviewed. Then, approaches used to model specific therapies are discussed. Following that, models of intracellular processes that are crucial in responses to therapies are presented. The paper is concluded with a discussion of the applicability of the presented approaches in the context of lung cancer.

## 1. Introduction

Despite decreasing trends concerning incidence and mortality in the male population, lung cancer is still one of the most commonly diagnosed cancers, the leading cause of cancer-related deaths [1], and annual cases are expected to reach 3.8 million in 2050 [2]. While 3-year relative survival increased to 31%, it is due not only to new treatments being developed, but also to the increase in localized-stage diagnoses [3]. Therefore, finding new and improving existing treatments for lung cancer is one of the most important challenges in oncology.

Various types of therapy, and protocols, have been approved for the treatment of lung cancer, and still more are either already in clinical trials or in different development stages. The choice of therapy mode depends on the cancer type, its molecular features specific for an individual patient, its grade, the general patient state, and other factors.

Standard cancer treatment protocols, approved by respective agencies, are developed based on carefully conducted trial studies. For the most part, either their parameters are rigidly defined, or there are only a few variants of protocols, concerning timing and dosage (see, e.g., [4]). Personalized medicine aims at changing that. Designing a treatment protocol for a trial study in new therapy tests is a much more complicated problem—and the trial success or failure may heavily depend on the protocol applied. Mathematical modeling may help in testing alternative protocols and estimate the variance in treatment results stemming from intra- and intertumor heterogeneity, thus providing valuable insight even before trials start. When personalized treatment is considered, the benefits of in silico tests are even greater.

This work is focused on models that have been developed to describe cancer growth and therapy effects. First, a short review of available therapies is presented, though it is limited and aimed only to provide basic background before introducing mathematical models. Those interested in details of therapies available, under development, or considered to be promising should refer to one of many medical reviews in the field (e.g., [5,6,7,8]). Then, a general concept of mathematical modeling applied to describe cancer at different levels of organization is introduced, followed by specific examples, relating to lung cancer and signaling pathways associated with it. The paper is concluded with remarks of applicability of the models developed so far and directions to be pursued in future research.

## 2. Lung Cancer Treatment and Associated Signaling Pathways

### 2.1. Treatment Options

The recommended treatments for lung cancer include, among others, surgery, radiotherapy, chemotherapy, various types of targeted therapy or immunotherapy, and a combination of these approaches. Details of the protocols are publicly available, e.g., [9,10,11]. Each of the therapy types has enormous literature devoted to it, dealing with efficacy, side effects, particular therapeutic agents, intracellular signaling, and regulatory networks involved in response to treatment, etc. This section lists the types of treatment that are considered in mathematical models described farther in the text.

Surgery is most often recommended for early-stage tumors. Though it may be regarded as a standard procedure, it is still a subject of ongoing research that is focused on determining the optimal resection area [12]. Computational methods are used for image analysis, used to support image segmentation and determine the tumor boundaries. Mathematical models of cancer growth and its response to treatment do not take it into account explicitly, and therefore, this type of treatment is not dealt with in this paper. Image-guided thermal ablation, also used in lung cancer treatment [13], is omitted for the same reason.

Chemotherapy remains to be another widely used option, for both NSCLC [14] and SCLC [15] subtypes. Though a combination of chemo- and immunotherapies (with a shift towards the latter) has been gaining increasing interest in recent years, chemotherapy is still regarded as a necessary treatment mode, needed even if targeted immunotherapy is available [16]. It is also used in palliative therapy, as increasing survival times due to reduced chemotherapy protocols have been reported, for treatment that does not involve very early palliation [17]. It is worth noting that the question of reduced dosage and its benefits have been the subject of research for many years in the context of metronomic chemotherapy [18]. However, even though new drug agents are developed to be used against lung (and other) cancer, chemotherapy efficacy is hampered by drug resistance that is evolved by cancer cells to escape treatment. There are many different mechanism of such resistance [19,20], and learning them requires gaining insight into their molecular context, thus opening new areas to mathematical modeling that supports molecular biology research.

Radiotherapy constitutes the third standard treatment type in fighting many cancer types, including lung cancer [21,22,23,24]. It comprises a large group of treatment options, including external beam radiotherapy, stereotactic body radiation therapy, brachytherapy, and stereotactic ablative brachytherapy. Of these, stereotactic body radiation therapy (SBRT) is the standard of care for inoperable early-stage [25] or oligometastatic NSCLC [26]. There is an ongoing debate whether stereotactic radiotherapy yields better results than other forms of treatment, with some papers reporting its advantage (e.g., [27,28]), while other claiming otherwise (e.g., [26,29]). It seems that the conclusions strongly depend on the characteristics of patient groups, and mathematical modeling might help in explaining these discrepancies. Sadly, comprehensive modeling studies comparing the effects of different radiotherapy options hardly exist.

As in chemotherapy, cancer cells may develop resistance to it [30], and similarly, learning molecular mechanisms behind it may lead to the development of new combined treatments of better efficacy [31].

Both chemo- and radiotherapy belong to the family of nontargeted therapies, as are less common options, such as radiofrequency ablation, microwave ablation, cryoablation [13], or photodynamic therapy. While the current trend is to develop targeted and personalized medicine, still much effort is directed toward increasing the effectiveness of existing agents [32] and limiting the damage of healthy cells through localizing the area of therapy actions [6].

In recent years, targeted therapies and immunotherapy have gained increasing popularity in the treatment of various types of cancer, including lung cancer [33,34]. A query for lung cancer immunotherapy in the PubMed database returns over 2300 articles published in the last year alone. The options range from antibody–drug conjugates (ADCs), PARP inhibitors, and tyrosine kinase inhibitors to immune checkpoint inhibitors, to name only a few. Additionally, immunotherapy is used together with other forms of treatment, including radiation [35]. Immunotherapy has also brought back antiangiogenic treatment, disregarded for some time as the one that did not fulfill its promise, as an important component in combined therapies [36,37].

### 2.2. Sample Molecular Players Involved in Tumor Response to Treatment

Molecular biology coupled with bioinformatics methods support oncology in two different aspects. First, they provide tools for finding new biomarkers of cancer of a specific type. Second, they help to unravel regulatory networks and signaling pathways that are behind carcinogenesis or cancer response to treatment, with respect to cancer metabolism, DNA damage detection, and repair mechanisms as well as overcoming cell cycle checkpoints [38]. That, in turn, facilitates the development of better treatment protocols and overcoming treatment resistance. This issue is particularly important in targeted and immunotherapies.

In the past, arguably most research efforts were directed at p53 and NF-κB associated pathways, as they are involved in processes activated by cells to minimize damage to important cellular targets caused by radio- or chemotherapy [39]. Jak/STAT pathways, utilized in many intracellular processes, are also investigated in the context of anticancer therapies [40]. Wnt and Notch signaling and EGFR were also investigated in much detail, as they are involved in epithelial–mesenchymal evolution, important for metastasis [41]. In the context of lung cancer, it is particularly important, as advanced NSCLC is likely to metastasize and the presence of distant metastases is one of the most predictive factors of poor prognosis [42]. However, other pathways, specific for particular resistance mechanisms, gained a lot of interests in recent years. Below, only a few examples are mentioned. An excellent thorough review can be found in [43].

Of the of molecular players involved in resistance to radio- or chemotherapy ABC family transporters, EGFR, MAPK, PI3K/Akt, PTEN, NRF2, BCL-2, and FOXF1 are the most often considered subjects of research [43,44]. More and more microRNAs are found to be responsible for radio- or chemoresistance as well as altering responses to immunotherapeutic actions [45,46]. This alteration can be also used for the benefit of anticancer therapies [47]. On the other hand, EGFR mentioned above in the context of therapy resistance is studied extensively as a major factor in immunotherapy using EGFR-tyrosine kinase inhibitors (TKIs) [48]. TLR is considered to be a convenient sensitization target in immunotherapy [49].

The brief outline given above shows an enormous scope of research needed to cover the area. However, much more work is needed to design alternative protocols that would take into account individual responses to treatment and determine synergistic or antagonistic effects with other forms of anticancer therapy. As these depend on intracellular processes activated during interaction between cancer and immune cells, the dynamics of these processes must be determined, and the heterogeneity of cellular responses must be taken into account when trying to predict the treatment outcome for specific protocols. Mathematical models allow for addressing these issues. Simulation provides convenient means to test alternative protocols, and analysis of stationary points of cell population models facilitates answering questions about conditions of eradication of cancer cells, containment of tumor size under a given threshold, relapse, etc. Additionally, models of signaling pathways associated with cancer cell responses to treatment may provide hints about yet-to-be-discovered mechanisms controlling them. Subsequent sections introduce mathematical modeling in the context of biomedical research summarized above.

## 3. Modeling Cancer Growth and Response to Treatment at Different Levels

The term model is ambiguous and, even in the field of cancer research, may have various meanings subject to the context it is used in. For experimentalists, an animal, in vitro, or a 3D model (e.g., [50]) of a disease means an organism or cell culture, studied to gain knowledge about progress or treatment responses of this disease in, e.g., a human. Statistical models are employed, e.g., to analyze high-throughput data or the survival of patients or generate artificial data if real-life data are not available. Graph models are used to represent structures and relations, like a structure of a regulatory network or binding site affinity. In this paper, models of dynamics of processes are considered, at either a cell population/tissue (cancer growth) or intracellular (signaling pathways, intracellular responses to treatment) levels. In this section, basic modeling concepts are introduced, without providing specific details or complex expressions that may appear in the equations.

### 3.1. Cancer Growth

Three main approaches are most often used to model tumor growth and its response to treatment: ordinary differential equations (ODEs), partial differential equations (PDEs), and agent-based methods (ABMs). The latter approach does facilitate only computational, simulation-based analysis, while the first two make it possible to use formal analytical methods that allow for determining some properties of the systems without making a priori assumptions about parameter values. Despite that, agent-based methods are increasingly popular, as the aforementioned advantage of ODEs and PDEs disappears for systems too complex to apply effectively analytical approaches.

ODE models describe dynamics of cancer growth in terms of tumor volume or density, or number of cells or concentrations of cells or molecular species. In some cases, variables representing cells or tissues other than cancerous are included, but their meaning is analogous. If $N\left(t\right)={\left[N_{1}\left(t\right),\ldots ,N_{R}\left(t\right)\right]}^{T}$ denotes a vector, representing types of cells taken into account in the model, $U\left(t\right)={\left[U_{1}\left(t\right),\ldots ,U_{M}\left(t\right)\right]}^{T}$ denotes a vector whose elements represent different therapy modes, and $X\left(t\right)={\left[X_{1}\left(t\right),\ldots ,X_{K}\left(t\right)\right]}^{T}$ represents a vector of other variables, e.g., concentration of nutrients, oxygen, and cytokines, all at time *t*, then the most general form of these equations is as follows:(1)$$\frac{dN}{dt}=f\left(N(t),X(t),U(t)\right)-g\left(N(t),U(t),X(t)\right)+h\left(N(t),X(t),U(t)\right),$$
(2)$$\frac{dX}{dt}=p\left(N(t),X(t),U(t)\right)-q\left(N(t),X(t)\right),$$
where $f\left(.\right)=[f_{1}\left(.\right),\ldots ,f_{R}\left(.\right)]$, $g\left(.\right)=[g_{1}\left(.\right),\ldots ,g_{R}\left(.\right)]$, $h\left(.\right)=[h_{1}\left(.\right),\ldots ,h_{R}\left(.\right)]$, $p\left(.\right)=[p_{1}\left(.\right),\ldots ,p_{K}\left(.\right)]$, and $q\left(.\right)=[q_{1}\left(.\right),\ldots ,q_{K}\left(.\right)]$ denote growth, therapy effect on cells, flow between compartments representing different cell types (if applicable), production, and degradation (or utilization), respectively. While the notation above is the most general, with functions f(.), g(.), p(.), and q(.) that may depend on all possible model variables, in particular models, any of them depends on just either one or two of N(t), U(t), and X(t). Different forms of these functions are considered, depending on model assumptions, therapy type, and variables taken into account (see, e.g., [51] for a general introduction to these models).

Functions $h_{i}\left(.\right)$, are usually scalar products of a vector of parameters $a_{i}$ and vector N(t):(3)$$h_{i}\left(N(t),U(t),X(t)\right)=a_{i}\cdot N\left(t\right)$$
with parameters that may depend on drugs, i.e., $a_{i}=a_{i}(U\left(t\right)$, e.g., in the case of chemotherapy involving chemostatic drugs or recruitment of quiescent cells back to the $G_{1}$ phase of the cell cycle.

Three basic forms of growth function are used. The simplest one is the exponential growth, which, under the simplifying assumption that the growth rate is not affected by nutrient availability or molecular concentrations of any molecule types in the cell neighborhood, or by therapy (i.e., therapy does not affect cell cycle), takes the following form:(4)$$f\left(N(t),U(t),X(t)\right)=\rho_{1}N\left(t\right),\;\;N\left(0\right)=N_{0}.$$

As the assumption about growth independence on therapy or nutrients is an oversimplification, the model can be changed so that growth rate is not constant $\rho =\rho \left(X(t),U(t)\right)$ (see [51] for such model variants). Alternatively, logistic or Gompertzian growth is assumed, which, in their simplest forms (once again, assuming growth independence on therapy or nutrients, with similar remarks as those given for exponential growth), are as follows, respectively:(5)$$f\left(N(t),U(t),X(t)\right)=\rho_{2}N\left(t\right)\left(1-\frac{N(t)}{K}\right),\;\;N\left(0\right)=N_{0},$$
(6)$$f\left(N(t),U(t),X(t)\right)=-\rho_{3}N\ln\frac{N}{N_{0}},\;\;N\left(0\right)=N_{0}.$$

In (Equation 4)–(Equation 6), $\rho_{i}$ and *K* represent growth rates and so-called carrying capacity (the latter limiting maximum population size) and are the model parameters.

Production functions in (Equation 2) are either explicit functions of time, if they represent an explicit input to the system (then, usually, $p_{i}\left(N(t),X(t),U(t)\right)=U_{i}\left(t\right)$), or usually given by linear or Michaelis–Menten–type expressions assuming that they are produced by cells in the population, i.e.,
(7)$$p_{i}\left(N(t),X(t)\right)=\sum_{i}k_{i}N_{i}\left(t\right),$$
or
(8)$$p_{i}\left(N(t),X(t)\right)=\sum_{i}\frac{k_{i1}N_{i}\left(t\right)}{k_{i2}+N_{i}\left(t\right)},$$
where $k_{i},k_{ij}$ are model parameters.

Depending on its interpretation, the function q(.) in (Equation 2) takes usually either a linear or bilinear form. If it represents natural degradation, then the linear term is used, i.e.,
(9)$$q_{i}\left(N(t),X(t)\right)=k_{deg\_i}X_{i}\left(t\right).$$

If the function q(.) represents utilization, or uptake, of molecules by cells in a population, then bilinear or Michaelis–Menten–type expressions are used, i.e.,
(10)$$q_{i}\left(N(t),X(t)\right)=\sum_{i}\sum_{j}k_{ij}N_{i}\left(t\right)X_{j}\left(t\right),$$
where $k_{i}$ are model parameters, or
(11)$$q_{i}\left(N(t),X(t)\right)=\sum_{i}\sum_{j}\frac{k_{ij1}N_{i}\left(t\right)X_{j}\left(t\right)}{k_{ij2}+X_{i}\left(t\right)}.$$

### 3.2. Therapy Modeling at the Cancer Cell Population Scale

#### 3.2.1. Surgery

As mentioned before, surgery itself is not modeled explicitly. If a model is to describe a case in which cancer grows first and then resection takes place, followed by some adjuvant therapy, it can be represented by a switch in $N_{i}$ value at surgery time, where the previous value is replaced by its fraction $\mu N_{i}\left(\mu \leq 1\right)$, sampled from a priori assumed distribution (μ=0 would mean not leaving a single tumor cell behind, and μ = 1—completely ineffective surgery). Otherwise, the analysis starts at t=0 representing the moment after resection, with an initial condition N(0) that represents tumor that remained after the procedure.

The same approach might be applied for modeling image-guided thermal ablation, which is another treatment.

#### 3.2.2. Chemotherapy

Chemotherapy effects are most often modeled following Skipper’s laws [52], stating that the relationship between dose and tumor regression is linear logarithmic. That leads to the following $g_{i}\left(.\right)$ form in (Equation 1), assuming a single relation between subpopulation *i* and the drug *j*, whose dose at time *t* is denoted by $U_{j}\left(t\right)$:(12)$$g_{i}\left(N(t),U(t),X(t)\right)=c_{ij}N_{i}\left(t\right)U_{j}\left(t\right),$$
or, in a more realistic case, when pharmacokinetics (PK) is taken into account, e.g., in its simplest form:(13)$$g_{i}\left(N(t),U(t),X(t)\right)=c_{ij}N_{i}\left(t\right)c_{j}\left(t\right),$$
where $c_{j}\left(t\right)$ denotes local concentration of drug affecting tumor, which is the solution to the following PK equation:(14)$$\frac{dc_{j}}{dt}=U_{j}\left(t\right)-kc_{j}\left(t\right),\;\;c_{j}\left(0\right)=0.$$

Despite many other approaches existing in the literature, the one given by (Equation 14), or its modifications that take into account pharmacodynamics (PD), shown below, seem to be prevalent.

When a combination of drugs is applied, the approach used depends on the drug types. If each of them targets cells in a different phase of the cell cycle; then the equations given above hold for compartments representing subpopulations in the specific phase and respective drug agent. If they may affect cells of the same type, their joint effect (additive or synergistic) must be considered. For example, in [53], the combined therapy effect for two drugs is described as a sum of their impact on the tumor cell population, but each drug’s effect takes into account a possible synergy between them:(15)$$g_{i}\left(N(t),U(t),X(t)\right)=N\left(t\right)S_{3\_1}\left(t\right)+N\left(t\right)S_{3\_2}\left(t\right)$$
where the individual drug’s effects $S_{3\_i}\left(t\right)$ are the solutions of the set of three ODEs, representing both PD and PK:(16)$$\frac{dS_{1\_i}}{dt}=\frac{1}{\tau_{i}}\cdot \left(\frac{K_{max\_i}c_{i}^{\gamma_{i}}\left(t\right)}{c_{i}^{\gamma_{i}}\left(t\right)+{\left(\Psi \cdot KC_{50\_i}\right)}^{\gamma_{i}}}-S_{1\_i}\right),$$
(17)$$\frac{dS_{2\_i}}{dt}=\frac{1}{\tau_{i}}\cdot \left(S_{1\_i}-S_{1\_i}\right),$$
(18)$$\frac{dS_{3\_i}}{dt}=\frac{1}{\tau_{i}}\cdot \left(S_{2\_i}-S_{3\_i}\right),$$
where $K_{max\_i}$, $c_{i}\left(t\right)$$KC_{50\_i}$, and $\gamma_{i}$ denote the maximum killing rate constant, the effective drug concentration in the tumor neighborhood, the concentration that induces 50% of the killing capacity, and the Hill coefficient, respectively, for the *i*-th drug. $S_{1\_j}$ and $S_{2\_j}$ (j=1,2,3) represent hypothetical signal transduction compartments for each drug that introduces a time delay in the downstream pharmacodynamic response, and $\tau_{j}$ denotes the mean transit time from one compartment to another. A possible sensitization to one drug by another drug’s action is represented by the Ψ parameter (for Ψ≤1, the drugs work in a synergistic way; for Ψ = 1, their effect is additive; and for Ψ≥1, they are antagonistic). A good review of drug synergy modeling can be found in [54].

#### 3.2.3. Radiotherapy

Radiation impact on cell population is usually taken into account in the form of the so-called linear quadratic (LQ) model, first formulated in [55] and still widely used today, either as originally devised or with some modifications. The cell loss rate in that model is given by the following relation [56]:(19)$$g_{i}\left(N(t),U(t),X(t)\right)=\left(\alpha d+\beta d^{2}\right)N,$$
with *d* representing irradiation dose and α, β being LQ model parameters. However, actual radiation doses, represented by the control variable U(t), are introduced into the model in a different way than in the modeling of chemotherapy protocols, where that variable represents an actual chemotherapy protocol. Instead of explicitly using U(t) as a series of impulses representing subsequent irradiation fractions, it is recalculated into the biologically effective dose (BED) [57]:(20)$$d=nd_{i}\left(1+\frac{d}{\alpha /\beta }\right)-\ln\left(\frac{2(T-T_{k})}{\alpha T_{p}}\right),$$
where *n*, $d_{i}$, *T*, $T_{p}$ denote the number of radiation fractions, a single dose in (Gy), the overall length of a radiation cycle, and tumor population doubling time in (days), respectively. It is assumed that the repopulation starts after $T_{k}$ days, the time delay needed to complete DNA damage repair, caused by the irradiation. Then, *d* is used in the model as a constant throughout the whole period of the radiation cycle. A brief review of modifications of that basic model can be found in [58].

The LQ model does not work well with high radiation doses, characteristic for stereotactic body radiation therapy. Among different approaches proposed to deal with that problem, the so-called microdosimetric kinetic model (MKM) [59] gained the largest following. In [60], it was merged with a compartmental ODE model, in which variables corresponded to active tumor cells, resting cells, and nondividing cells. The dynamics of the subpopulations were modeled similarly as for the LQ model, described above. However, the parameters corresponding to the radiation-induced death were not constant but calculated using the MKM approach.

It should be noted that mathematical modeling supporting radiotherapy is a much broader topic than indicated by the considerations given in this section. For example, methods developed for image analysis focused on either diagnostics or beam-guidance purposes [61], or more detailed dosimetric models [62] have been omitted, as they would call for a separate review paper.

#### 3.2.4. Antiangiogenic Treatment

The modeling of the antiangiogenic treatment requires taking into account vasculature growth, in addition to cancer growth. To keep notation consistency throughout this paper, let this be described by $N_{2}\left(t\right)$, with $N_{1}\left(t\right)$ describing cancer volume in the general vector $N\left(t\right)=[N_{1}\left(t\right)N2\left(t\right)]$. Then, the simplest model would include the following growth functions [63]:(21)$$f_{1}\left(N(t),U(t),X(t)\right)=-\rho_{3}N_{1}\ln\frac{N_{1}}{N_{2}},\;\;N_{1}\left(0\right)=N_{10},$$
(22)$$f_{2}\left(N(t),U(t),X(t)\right)=k_{1}N_{1}-k_{2}N_{2}N_{1}^{2/3},\;\;N2\left(0\right)=N_{20}.$$

The function $f_{1}\left(.\right)$ comes from the application of the Gompertz growth given by (Equation 6) but with the maximum tumor size limited by the vasculature $N_{2}\left(t\right)$ available. The function $f_{2}\left(.\right)$ accounts for the release of vasculature growth factors (first term) and inhibitors (second term) by tumor cells. Since the antiangiogenic drugs do not affect cancer cells directly, the therapy effect given by $g_{1}\left(N(t),U(t),X(t)\right)=0$. The impact of the antiangiogenic treatment on the vasculature is calculated in a way similar to the chemotherapy effect in (Equation 12):(23)$$g_{2}\left(N(t),U(t),X(t)\right)=c_{2}N_{2}\left(t\right)U\left(t\right),$$
where U(t) represents antiangiogenic drug concentration.

There are many versions of this model, utilizing logistic instead of Gompertz cancer growth or slightly different expressions representing tumor-induced vasculature growth (see their analysis in [64]).

#### 3.2.5. Immunotherapy Models

Due to the increasing interest in immunotherapies, many mathematical models have been developed recently in addition to those tumor–immune interactions proposed much earlier. New models usually describe combined therapies, where immunotherapy is one of the modalities used. The simplest model of tumor–immune interactions is based on two variables, $N_{1}$ and $N_{2}$, representing tumor and immune effector cells, respectively (once again, the original notation is changed here to maintain consistency with the one used throughout this paper), and is based on the predator (immune cells)–prey(tumor) models. The tumor growth function can be any of (Equation 4)–(Equation 6), and the loss of tumor cells due to their interactions with immune effector cells is given by
(24)$$g_{1}\left(N(t),U(t),X(t)\right)=k_{1}N_{1}\left(t\right)N_{2}\left(t\right),$$
with $k_{1}$ being the model parameter. Since the effector cells are supposed to be activated by tumor cells, their growth rate function is given by
(25)$$f_{2}\left(N(t),U(t),X(t)\right)=k_{2}N_{1}\left(t\right).$$

Introducing therapy in that model is straightforward—if it is based on the injection of immune effector cells, then an additional term U(t) is added in the above equation, representing it (e.g., in [65]).
(26)$$f_{2}\left(N(t),U(t),X(t)\right)=k_{2}N_{1}\left(t\right)+U\left(t\right).$$

The loss rate of the immune effector cells is either assumed to be linear $g_{2}\left(N(t),U(t),X(t)\right)=p_{1}N_{2}\left(t\right)$ or takes into account an additional term, describing their utilization in the interaction process [66]:(27)$$g_{2}\left(N(t),U(t),X(t)\right)=p_{1}N_{2}\left(t\right)+p_{2}N_{1}\left(t\right)N_{2}\left(t\right).$$

Other models take into account rate-limited tumor–immune interactions, resulting in the loss of tumor cells given by [67]
(28)$$g_{1}\left(N(t),U(t),X(t)\right)=k_{1}\left(\frac{N_{2}/N_{1}}{k_{2}+N_{2}/N_{1}}\right)N_{1}\left(t\right).$$

More complex models distinguish different types of immune cells [68], separate compartments, and additional phenomena such as time delays, or include signaling molecules that mediate tumor–immune interactions, represented by separate model variables X(t), leading to models in the form given by (Equation 1) and (Equation 2). This, in turn, leads to population-level models of targeted therapies like in [69], where the vector X(t) of concentrations of molecules taken into account consists of four components, representing free fibroblast growth factor receptors, their active dimer complexes, programmed cell death protein 1 (PD-1), and programmed death-ligand 1 (PD-L1). Production and degradation rates in (Equation 2) are nonlinear functions. For example, if X−1(t) and $X_{2}\left(t\right)$ denoted concentrations of free fibroblast growth factor receptors and their active dimer complexes, respectively, then corresponding functions in (Equation 2) would take the following form (for more details, and remaining functions, see [69]):(29)$$p_{1}\left(N(t),U(t),X(t)\right)=k_{1}X_{1}^{2}+k_{2}X_{2}+k_{3}N_{1},$$
(30)$$q_{1}\left(N(t),U(t),X(t)\right)=\frac{c_{1}N_{2}}{1+c_{2}X_{2}/N_{1}}.$$

As that model accounts for programmed death-ligand 1 (PD-L1) blockade, it includes a component of targeted therapies, in which specific molecules are used to activate or inhibit specific molecular processes. Since targeted therapies involve taking advantage of knowledge of intracellular signaling pathways and regulatory networks, they employ models briefly described in the subsequent sections.

A good review of immune and targeted therapy models can be found in [70].

#### 3.2.6. Models of Combined Therapies

In most cases, models of combined therapies are created by merging models of their components. For example, when radiochemotherapy is considered, the therapy effect, described in (Equation 1) by the function g(.) is a sum of terms given by (Equation 12) and (Equation 19), i.e., [71]:(31)$$g_{i}\left(N(t),U(t),X(t)\right)=k_{1}c\left(t\right)N\left(t\right)+k_{2}\left(\alpha d+\beta d^{2}\right)N\left(t\right),$$
where $U\left(t\right)=[U_{1}\left(t\right)d]$, representing chemo- and radiotherapy components, with c(t) related to $U_{1}$ dose by the PK model (Equation 13) and *d* being the biologically effective dose defined by (Equation 20).

For chemoimmunotherapy, a separate variable representing drug concentration is added, together with PK equations and respective terms describing the chemotherapy effects, exactly as in a pure chemotherapy model [72,73]. The same is true for combined radio- and immunotherapy [74,75].

Model (Equation 31) does not take into account synergistic or antagonistic effects of one therapy on another in combined therapies. Though some papers claim that a possible interaction between chemo- and radiotherapy may be neglected [71], other works account for a chemotherapy-induced sensitization of cells to concurrent radiation, e.g., through the modification of the LQ model parameters [76]. Other modeling works focus on other sensitization approaches, e.g., thermal radiosensitization [77].

#### 3.2.7. Therapy Resistance and Metastasis

Therapy resistance is usually incorporated in the model by introducing separate variables, representing sensitive and resistant (or partially resistant) cells, and differentiating therapy functions in respective equations. The modification of the g(.) function usually consists in multiplying the function that represents tumor cell loss for a sensitive population by a factor γ∈[01], where the smaller γ is, the stronger therapy resistance is exhibited. Such approach is used in models dealing with any therapy type, describing chemoresistance or radioresistance or resistance to immune checkpoint inhibitors in lung cancer [78].

While the resistance was quite often analyzed in the context of a single type of therapy, nowadays, it is dealt with using an alternative modality meant to overcome it (e.g., antiangiogenic and chemotherapy [79]), leading to the models of combined therapies. Other approaches are based on the analysis of intracellular regulatory pathways and specific signaling pathways involved in therapy resistance mechanisms [80], and the models used to deal with it are quite often multiscale, agent-based or hybrid models mentioned farther in the text.

Similarly, metastasis is quite often incorporated by adding separate compartments representing these sites, and flows from primary to metastatic compartments, thus fitting into the general model given by (Equation 1) and (Equation 2) [81]. Parameters for metastatic compartments are usually different from those for primary tumor, accounting for their partial or total therapy resistance or more aggressive growth. Another approach might involve using partial differential equations, as described in the subsequent section.

### 3.3. PDE Models

The main drawback of ODE models is not taking into account the heterogeneity of tumor cells with respect to their cell cycle length and responses to treatment. To some limited extent, PDE models may help overcome it. Using a PDE-based approach makes it possible to take into account spatial effects in addition to temporal ones, cell age structure, different levels of therapy resistance, and even metastasis.

In majority of cases, these models are built upon reaction–diffusion–chemotaxis equations [82]:(32)$$\frac{\partial N}{\partial t}=f\left(N\right)-\nabla \cdot \left(N\chi (X)\nabla X\right)+\nabla \cdot D\nabla N,$$
(33)$$\frac{\partial X}{\partial t}=p\left(X,N\right)+\nabla \cdot D_{X}\nabla X.$$
where $N_{\chi }\left(X\right)$ denotes a function of attractant concentration that drives the chemotaxis, while *D* and $D_{X}$ denote diffusion coefficients for cells and attractant, respectively.

Drug resistance can be taken into account in a PDE modeling framework by associating resistance with a continuous variable x∈[0,1], with x=0 representing a sensitive phenotype, while *x* = 1—a totally resistant one. Then, denoting by N(t,x) the density of cells at time *t* and with phenotype *x*, the population dynamics can be described by the equation similar to (Equation 1), but in a PDE form [83]:(34)$$\frac{\partial N}{\partial t}=f\left(N(t),U(t),X(t)\right)-g\left(N(t),U(t),X(t)\right),$$
where the loss function g() depends on a phenotype in a manner given by a function d(.) (which may take different forms, subject to model assumptions):(35)g=d(x,U(t))ρ(t)N(t,x)
and
(36)$$\rho \left(t\right)=\int_{0}^{1}N\left(t,x\right)dx$$

An interesting application of PDE can be found in [84], where the number of metastatic sites is not an integer but a continuous variable ρ(x,t) (colony size distribution), denoting a definite number of metastatic tumors of size from *x* to x+dx. Its changes in time are described by the following PDE:(37)$$\frac{\partial \rho (x,t)}{\partial t}+\frac{\partial g(x)\rho (x,t)}{\partial x}=0$$
with initial and boundary conditions given by
(38)$$\begin{array}{ccc} \rho (0,x) & = & 0 \end{array}$$
(39)$$\begin{array}{ccc} g(1)\rho (1,t) & = & \int_{1}^{\infty }\beta \left(x\right)\rho \left(x,t\right)dx+\beta \left(x_{p}\left(t\right)\right). \end{array}$$

In [84], the growth rate g(x) was assumed to be a Gompertzian, but depending on model assumptions, it could be any of the functions given by (Equation 4)–(Equation 6).

The emission rate β(x) was assumed to take the following form:(40)$$\beta \left(x\right)=mx^{\alpha }$$

The growth of primary tumor $x_{p}\left(t\right)$ can be described by the general form given by (Equation 1), though in the original work [84], no therapy-related component was given.

The examples given below show that the PDE approach makes it possible to overcome some drawbacks of ODE modeling, by allowing for representing the heterogeneity of cancer cells and spatial effects. However, it is still a deterministic modeling framework and, thus, cannot capture stochastic phenomena that arise from tumor growth and treatment. To overcome that, agent-based models (or hybrid models) are used.

### 3.4. Agent-Based Models and Multiscale Modeling

Agent-based modeling (ABM) is a computational approach that uses the so-called agents representing individual cells or cell subpopulations, usually distributed on a 2D or 3D spatial grid. Model rules determine interactions between them and between agents and molecular species that might be also present in the grid (such as oxygen and nutrients), affecting agents’ discrete states (e.g., proliferation, death, release of communication signals) [85]. The transport and utilization of theses molecular species are modeled either by separate agents or by means of PDE in space superimposed on the grid. Since each agent may behave in a different, stochastic-driven way, the heterogeneity of cancer cells is naturally represented, which is one of the most important advantages of this approach.

To illustrate the concept of the ABM approach, let us consider a relatively simple example of 2D tumor growth that depends on oxygen, nutrients, growth factors, growth inhibitors, and killing agents that happen to be in its neighborhood. Let each cell grid represent a small homogenic tumor cell subpopulation (an agent) that can be in one of several states, e.g., proliferating, with a stopped cell cycle and trying to repair DNA damage, senescent, necrotic, apoptotic, or empty. Additionally, it might be described by a vector of additional features, such as the cell cycle phase and density (to be used by rules that are used to decide if cells in the subpopulation divide, thus increasing the agent density), or therapy resistance level. The agent can change the state subject to a set of rules that define the influence of neighboring agents and concentrations of all factors that affect it (for a neighborhood that needs to be defined—Figure 1). To take the latter into account, a separate agent grid is created, in which each agent represents a vector of local concentrations of these factors. A separate list of rules needs to be defined to describe how the concentrations change, i.e., how the state of each agent in the second grid changes. These rules may describe, e.g., diffusion-led concentration changes, or the release or uptake of molecules by the agents in the first grid. Initial conditions define the state of each agent in both grids. Additionally, border conditions may be used to not only introduce the necessary parameters for running a simulation, but also introduce a therapy-related input, e.g., chemotherapeutic agents or immune cells.

Once all rules and parameters have been defined, a simulation starts, showing spatiotemporal tumor growth and its response to treatment. There are various procedures of updating the agents’ states (synchronous and asynchronous constitute an example of possible approaches). The simplest one would involve calculating a new state for each agent in the first grid, starting from the upper left corner and moving to the right and then to the next row, which would be followed by an update of the second grid. Snapshots of the grid in a simulation may be then compared with medical image data, such as CT or MRI scans. To compare this approach with those based on differential equations, the total number of cells in respective subpopulations must be calculated from the densities characterizing the agents. The length of a simulation is chosen arbitrarily, or the simulation is run until the steady state is reached (i.e., the first grid does not significantly change from one iteration to another). This includes also the case when the grid is composed of the same tumor cells, since the grid size does not allow the tumor to grow further, or with dead cells.

It should be noted, that at least some of the rules of state change are probabilistic, which means that one needs to run multiple simulations to be able to draw meaningful conclusions.

If the agents represent individual cells and their behavior is determined by rules based on the description of the dynamics of intracellular processes, they are often referred to as hybrid or multiscale models. Such models enable the prediction of tumor growth under given molecular properties, microenvironment conditions, and drug PK/PD profile. Various frameworks have been developed in this area, including discrete dynamic network models [86].

ABM or hybrid models found multiple applications in modeling cancer growth, cancer vascularization, and its response to treatment, including chemotherapy [87], radiotherapy [88], antiangiogenic [89], immunotherapy [90,91], to name just a few examples.

Despite the popularity of these methods, their two main drawbacks should be stated: lack of analytical methods for a qualitative analysis of such models and large computational burden for large grids.

### 3.5. Changing the Perspective from a Tumor in a Single Patient to a Patient Population

It should be emphasized that any of the approaches described above refer to modeling cancer growth and response to treatment for a single patient. Tuning parameters for these models constitutes a very difficult and delicate task due to lack of adequate, dense measurement data. Bioimaging data are usually available (if they are available at all) for two time points (before and after treatment). Though there is a lot of research into biomarkers that could provide information about the state of the disease, it is not possible yet to use them to determine the size of the tumor. Moreover, clinicians are used to evaluating treatment efficacy based on Kaplan–Meier curves, showing either overall survival or metastasis-free survival in a population of patients. Therefore, models should be able to take into account intertumor heterogeneity in a population of patients. This is usually achieved by the creation of a pool of virtual patients that are described by the same model but with different parameters [92]. Such setup makes it possible to computationally test the efficacy of alternative treatment protocols with either a standard protocol used for all patients, and evaluate its quality by means of Kaplan–Meier curves, or a personalized treatment for each patient and look at the patient’s response [93].

## 4. Modeling Intracellular Processes Associated with Cancer Growth and Its Responses to Treatment

### 4.1. General Modeling Remarks

While various approaches were developed to model intracellular processes [94], ODE (or PDE, if spatial phenomena are to be captured) models seem to be the most popular. Variables represent concentrations, levels, or the number of molecules of molecular species that are taken into account in a given system. The processes taken into account include translation, transcription, degradation of proteins, transcripts and other molecules, production of other molecules (such ROS), complex creation and dissociation, and degradation of molecules. Since it is impossible to take into account interactions between all possible types of molecules, it is assumed that the system under consideration is not affected by molecules and interactions not included in the model (which usually is an oversimplification). As a result, the models developed describe individual regulatory networks or regulatory pathways that are important from the research focus (e.g., cancer treatment or cell cycle or immune response). When the ODE modeling framework is used, the nonlinear state equation that is used can be represented by the form of (Equation 2). Such models help to find their properties in terms of transient responses and stationary states.

Sometimes, such approaches are expanded by adding a component from other methodologies, e.g., fuzzy logic [95], Petri nets, [96] or others [97].

Knowledge gained from the computational analysis of these models is key to understand possible types of system behavior and provides valuable information concerning, e.g., the planning of molecular biology experiments aimed at confirming hypotheses about control mechanisms governing cellular responses to treatment. Moreover, the computational analysis of these models makes it possible to test certain hypotheses that would be impossible to check experimentally, e.g., due to lack of available antibodies, sequences, or insufficient current biological knowledge (about intermediary proteins or protein complexes involved). Properly constructed models help to elucidate inconsistencies in the experimental results that might be the result of the heterogeneity of cellular responses (see, e.g., [98,99]). This is particularly important when the molecular players may promote or inhibit certain processes, depending on the cellular state (such as NF-κB pro- or antiapoptotic actions [100]).

Models of intracellular processes may also be used in the search for prospective new drug targets [101], which, when combined with structural biology and molecular dynamics analysis, significantly reduces the time needed to design new drugs.

Finally, as the intracellular responses determine cell fate and intercellular signaling, these models can also be used as a component of a more general framework, describing responses of tumor to therapeutic actions, with cell death, change of proliferation rate, intercellular signaling, or interactions with immune cells resulting from individual cell responses.

### 4.2. Examples of Models of Intracellular Processes Related to Cancer Growth and Its Response to Therapy

Arguably, regulatory networks most important for the analysis of cellular responses to antitumor therapy are those determining cell fate in general and cell death in particular. While it may seem that the scope of research is narrowed that way, it is unfortunately not true, as even when considering cell death, one must take into account the various processes leading to it, such as apoptosis, necroptosis, and ferroptosis [102], each associated with a different regulatory network.

Signaling pathways and regulatory networks, associated with cancer progression, metastasis, response to treatment, and therapy resistance have been the subject of modeling investigations. Taking into account their impact on cell population dynamics and the prospective development of multiscale models, facilitating personalized medicine progress, models of the following pathways, or regulatory modules seem to be the most promising (only sample references are given in the list below):p53 regulatory network [103,104];JAK/STAT signaling pathway [99,105];NF-κB [98,106];Large systems involved in DNA damage detection and repair, involving ATM, PTEN, and p53 proteins [96];NRF2 pathway [107];Wnt pathway [108];Cell cycle models [109].

Despite a lot of efforts in the area, these models have hardly been used to build multiscale models and thus facilitate the development of new treatment protocols. Moreover, models of intracellular processes are usually built upon in vitro experiments involving cell lines, and thus, the tissue context is not taken into account. That has begun to change only recently, with the development of 3D biological models.

## 5. Discussion

While various models have been developed and used for the analysis of cancer growth and treatment, relatively few of them are focused specifically on lung cancer. More precisely, even models whose parameters were fitted to clinical data of lung cancer patients do not take into account phenomena that are specific for lung cancer only. They utilize general modeling techniques and frameworks, changing parameter values, if necessary, to produce either transient responses to treatment or survival curves or particular indices such as metastasis-free survival that are observed clinically. The models were developed that way for both SCLC [110] and NSCLC [68], as well as Lewis lung carcinoma xenografts grown in immunogenic mice [111]. One of the indirect proofs that such approach works can be seen in the context of metronomic chemotherapy, both curative [112] and palliative [17], and combined with other forms of treatment [79], which have recently been tested through modeling in the lung cancer context [113]. One of the most important issues in cancer treatment—its adverse effects—has also been modeled for NSCLC [114].

Time scales used in simulations vary, depending on the type of the model. Generally, models at the population level cover months, if the goal is to observe tumor growth rate and direct therapy effects, to years (usually 2 or 5), if the treatment response is modeled for a virtual patient to find survival curves. For models of signaling pathways, simulation time usually does not exceed several hours, since additional molecular mechanisms are switched on as time passes and become impossible to neglect in the model structure, increasing the complexity of the model and making analysis very difficult, if possible at all. Moreover, model parameters are fitted to experimental data, and experiments’ duration is most often constrained to anything between 1 and 12 h, with a much smaller fraction of experiments looking at what happens after 24 or 48 h.

As far as spatial scales are concerned, two approaches prevail. The first one relates to existing imaging data, which makes it possible to estimate tumor size. Then, the spatial scale is directly related to what is observed in data. In the second approach, spatial dimension depends on the computational burden involved in simulation. Then, more often than not, that scale is not biologically relevant and the models are used to draw rough conclusions only.

It seems that the models that facilitate a comparison of alternative treatment protocols have been the most useful for some time now. In [115], a simulation-based comparison of alternative radiotherapy protocols for NSCLC allowed for concluding that two dose fractionation schedules, continuous hyperfractionated accelerated radiotherapy that included weekend irradiation and hyperfractionated accelerated radiotherapy weekend, yield almost the same long term-effects on locoregional NSCLC tumor control. While there were clinical trials devoted to the analysis of weekend gaps in head and neck tumors [116], no such studies were performed for lung cancer. That way, modeling work provided additional argument in the debate about weekend gaps in radiotherapy that are standard in health care systems.

Of the many papers devoted to the modeling of metronomic chemotherapy, one should distinguish [117], which proposed an alternative, counterintuitive vinorelbine dosage protocol for NSCLC, based on a computational study that could not have have been identified simply by analyzing the results of reported clinical trials. It is one of a few cases, when modeling was actually followed by a clinical trial that tested its applicability, confirming that the suggested protocol yields better efficacy [118]. Another study, combining mathematical modeling with in vivo experiments, showed that chemotherapy with doses smaller than MTD in NSCLC treatment leads to longer survival times and a lower level of drug resistance. It was also proven in the modeling of combined chemo- and antiangiogenic therapy [79], that the model was a general one, not specific for lung cancer. The mathematical analysis used in these works makes it possible to generalize the properties of the treatment under consideration and distinguish those that are parameter independent from properties arising under specific conditions only. Such conclusions could not be reached in studies based on clinical results only.

Sensitivity analysis of the models shows the most promising ways to increase treatment efficacy. The model developed in [68] showed somehow an intuitive result that it can be achieved in immunotherapy by increasing macrophage repolarization from the protumor anti-inflammatory M2 cells to the antitumor proinflammatory M1 cells. Less intuitive was another important conclusion that single immunotherapies might not affect tumor significantly. Only combining two immunotherapy approaches could be effective, even leading to complete tumor elimination (though the latter might require unfeasible biological parameters).

As mentioned earlier in the text, simulation models are particularly valuable in modeling immunotherapy, and this has been recently addressed in the context of lung cancer showing how possible treatment protocols can be simulated to identify possible problems that may arise [119]. Analysis of neoadjuvant PD-1 inhibition in NSCLC was studied in [120]. Important implications of immune-mediated metastatic growth on metastatic dormancy, blow-up, early detection, and treatment have been stated in [121], following a careful model analysis. It is worth noting that interactions between lung cancer, primary and metastatic, and the immune system have been studied in silico for many years now, also in the context of lung cancer immunotherapy [122,123].

Relatively few works have been published on modeling therapy adverse effects. However, this is slowly beginning to change, as quantified data on these effects start to appear, also in the case of adverse effects in lung-cancer-targeted therapies [124]. In [114], mathematical models led to the formulation of guidelines concerning adjuvant steroid protocols that are used to support targeted therapy of metastatic lung cancer.

The mathematical modeling of signaling pathways or regulatory network dynamics that can be associated with either a tumor response to treatment or mechanisms of its evasion has been developed in recent years (e.g., [105]), but it has been hardly utilized in higher-level analysis. Similarly, as in population-level modeling, there is an open question about the similarity in dynamics and structure of intracellular pathway models developed for other cancer types (particularly p53, NF-κB, Wnt, or NRF2 pathway models, important for chemoresistance) and their analogues in lung cancer.

Taking all of the above into account, current challenges in modeling lung cancer growth and treatment include the following:Trying to fit models into data for lung cancer in immunotherapy [15,125];Transforming structure, graph models of lung cancer metastasis, or regulatory networks involved in cellular responses to treatment into models describing the dynamics of these processes;Linking models describing synergistic effects of drugs in chemotherapy with models of cancer growth;Development of multiscale models, linking intracellular dynamics, specific for lung cancer, with models of metastasis or therapy resistance.

Achieving the goals stated above requires a lot of effort and cooperation of researchers whose interests are focused on model development, mathematical analysis, experimental work, and clinical data analysis. Therefore, in the near future, one should expect only gradual progress and the impact on clinical practice coming mostly from population-level modeling.

## Figures and Tables

**Figure 1 ijms-24-14516-f001:**
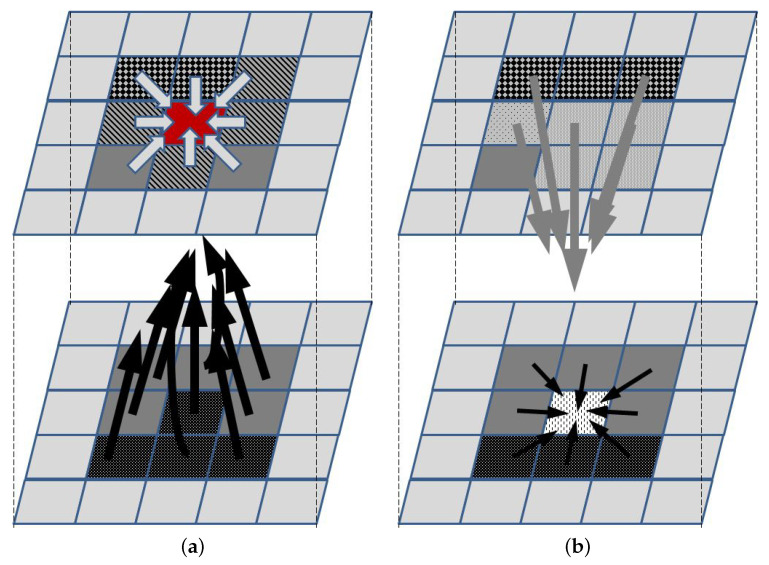
Main concept of ABM. The upper and lower panels represent the agents and the supplementary grids, respectively. (**a**) The agent represented by the red element changes its state according to the set of rules that define interactions with neighboring agents (light arrows) and the impact of elements whose concentrations define the state in the supplementary grid (black arrows). Once the new state for each agent is defined, then (**b**) concentrations in the supplementary grid are updated, according to the rules defining the impact of agents in the neighborhood (gray arrows) and concentrations in the neighborhood (black arrows).

## Data Availability

Not applicable.

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
