# Peer review of "Mathematical Modeling Support for Lung Cancer Therapy—A Short Review"

_ijms, 2023, doi:10.3390/ijms241914516_

Round 1

Reviewer 1 Report

This is an interesting review manuscript. It is well structured and good written. However, there are several points the authors need to carefully address in order to improve their paper. Most notably, they should provide some argumentation about the mathematical modelling papers they cite, and they should present as part of the narrative the new knowledge and understanding (towards building even better predictive models).

There no mention about stereotactic radio-surgery to treat lung cancer. Despite the authors mention about the LQ model, they do not discuss its appropriateness for lung cancer RT, the presentation of the models in this topic is very superficial. For example there is no presentation about surrogate models on image/model-guided radiation treatment.

There is no distinction in this paper between the models about primary lung cancer treatment to the models pertinent to simulate treatment of metastastic lung carcinomas.

In addition, there is no separate presentation of the mathematical (and computational models) for lung cancer simulations with respect to the termporal and spatial scales they describe, as well as the resolution they include. Also the review does not argue about the integration of clinical, or even preclinical, data to the modelling, thus, there is no information in the manuscript about the technological gaps in the field. The authors should therefore give a solid description how mathematical models have been informed and validated against real world data (or from preclinical models if that is the case). 

The list of papers cited is very long but still this review manuscript lacks in focus. For example, there are numerous papers that are irrelevant to lung cancer treatment (clinical, biology, mathematics, etc.) but they are about prostate, or brain, or melanoma, or breast, while there are several papers that are too outdated and do not give a fresh view to the reader about the new developments in the field.

A few not so major comments as well:

Some equations are difficult to read - the authors need to revise their notation as well as fix formating issues (e.g., what's the reverse hat on equations (1) and (2)?) or unnessary typos (e.g., double == in line 157) or peculiar text appearing as in equation (12), and should clarify symbols used in equations. Authors must thus do a careful review/screening of their manuscript before resubmission!

Not sure if this is the journal format, but reference numbers are mixed and are not sorted in order of appearance in the text. Can this be fixed please? Also, what are references 146-148, typos?

The authors are advised to consult for support from a professional to improve the language in their manuscript.

check above

Reviewer 2 Report

Author reviewed methods of mathematical modeling applied to lung cancer therapy. Three main approaches were described, including models based on ordinary and partial differential equations and agent-based models. Both the intracellular and population-level aspects were covered. The manuscript is well written, except typos, and a wide range of publications is considered.

I have the following questions/remarks:

1.     The author mostly focuses on the formal aspect of the modeling, well establishing the general form of the models first and then refining details depending on the context. However, it’s left unclear how the presented models really ‘support’ the therapy (citing the title). The general idea of why models can be useful is clear and well explained in the text, but I think the text is missing examples of fruitful insights or interesting outcomes that specific models of a given type produced regarding biological processes considered by those models. The manuscript would significantly benefit if the author added such examples for each modeling approach.

2.     The agent-based modeling framework description seems too short. Probably, a figure illustrating the ‘mechanics’ of the approach in the context of modeling lung cancer therapy and/or more detailed description of any specific study would help a lot.

The text contains multiple typos and math uncertainties. Here are some of them:

3.     About N in line 155: Types of cells are represented by subscript i of N_i, while actual N_i’s represent a number/density of cells of type i? Also, the max value of the subscript is also N, which is confusing.

4.     Equations (1) and (2) contain an undefined symbol instead of multiplication sign. Is it a typo or any specific math operator?

5.     Equation (12): not clear.

6.     Equation (14): should be no ‘t’ in the initial condition.

7.     Equation (20): d_i should be instead of d in the right part of this formula?

8.     Equation (32): What is N_chi(X)? Also, D and D_X should be formally introduced in the text after the equation.

I find quality of English good, but multiple typos need attention.

Round 2

Reviewer 2 Report

I'm satisfied with the corrections.